# Association of HbA_1C_ Variability and Renal Progression in Patients with Type 2 Diabetes with Chronic Kidney Disease Stages 3–4

**DOI:** 10.3390/ijms19124116

**Published:** 2018-12-18

**Authors:** Mei-Yueh Lee, Jiun-Chi Huang, Szu-Chia Chen, Hsin-Ying Clair Chiou, Pei-Yu Wu

**Affiliations:** 1Division of Endocrinology and Metabolism, Department of Internal Medicine, Kaohsiung Medical University Hospital, Kaohsiung Medical University, Kaohsiung 807, Taiwan; lovellelee@hotmail.com (M.-Y.L.); phoenixchiou@gmail.com (H.-Y.C.C.); 2Graduate Institute of Clinical Medicine, College of Medicine, Kaohsiung Medical University, Kaohsiung 807, Taiwan; karajan77@gmail.com (J.-C.H.); scarchenone@yahoo.com.tw (S.-C.C.); 3Faculty of Medicine, College of Medicine, Kaohsiung Medical University, Kaohsiung 807, Taiwan; 4Division of Nephrology, Department of Internal Medicine, Kaohsiung Medical University Hospital, Kaohsiung Medical University, Kaohsiung 807, Taiwan; 5Department of Internal Medicine, Kaohsiung Municipal Hsiao-Kang Hospital, Kaohsiung Medical University, Kaohsiung 812, Taiwan

**Keywords:** HbA_1C_ variability, chronic kidney disease, end-stage renal disease

## Abstract

Little is known about the predictive value of glycosylated hemoglobin (HbA_1C_) variability in patients with advanced chronic kidney disease (CKD). The aim of this study was to investigate whether HbA_1C_ variability is associated with progression to end-stage renal disease in diabetic patients with stages 3–5 CKD, and whether different stages of CKD affect these associations. Three hundred and eighty-eight patients with diabetes and stages 3–5 CKD were enrolled in this longitudinal study. Intra-individual HbA_1C_ variability was defined as the standard deviation (SD) of HbA_1C_, and the renal endpoint was defined as commencing dialysis. The results indicated that, during a median follow-up period of 3.5 years, 108 patients started dialysis. Adjusted Cox analysis showed an association between the highest tertile of HbA_1C_ SD (tertile 3 vs. tertile 1) and a lower risk of the renal endpoint (hazard ratio = 0.175; 95% confidence interval = 0.059–0.518; *p* = 0.002) in the patients with an HbA_1C_ level ≥ 7% and stages 3–4 CKD, but not in stage 5 CKD. Further subgroup analysis showed that the highest two tertiles of HbA_1C_ SD were associated with a lower risk of the renal endpoint in the group with a decreasing trend of HbA_1C_. Our results demonstrated that greater HbA_1C_ variability and a decreasing trend of HbA_1C_, which may be related to intensive diabetes control, was associated with a lower risk of progression to dialysis in the patients with stages 3–4 CKD and poor glycemic control (HbA1c ≥ 7%).

## 1. Introduction

Diabetes mellitus (DM) is the leading cause of chronic kidney disease (CKD) worldwide including Taiwan, where it has been reported to account for approximately 45% of all cases of end-stage renal disease (ESRD) in patients undergoing dialysis. Glucose control has been reported to be an important factor in controlling diabetic nephropathy, and time-averaged mean levels of glycemia, as assessed by glycosylated hemoglobin (HbA_1C_) level, is the gold standard of treatment to control glycemia and reduce the complications associated with diabetes [1,2]. Current guidelines suggest a target level of HbA_1C_ of approximately 7% to prevent or delay the microvascular complications associated with diabetes, including kidney disease [3,4]. However, several randomized controlled trials have reported that lowering blood glucose did not appreciably reduce the incidence ESRD or stop the progression of renal function [5,6,7]. This may be because these studies used HbA_1C_ to evaluate glucose control, which did not reflect glucose variability or the risks associated with extreme changes in glucose level over a long period of time [8].

Glycemic variability (GV) means swings in blood glucose level. Diminished or absent glycemic auto regulation or short falls of insulin availability are hypothesized to be the etiological factors for these glycemic bumps. The broad definition of GV takes into account the intraday glycemic excursions including episodes of hyper and hypoglycemia. The postprandial hyperglycemic excursions also contribute to GV [9]. GV has been shown to be positively associated with the development of diabetic retinopathy, cardiovascular events, and mortality in subjects with type 2 DM and preserved renal function [10]. In addition, with regards to nephropathy, HbA_1C_ variability has also been reported to be a significant predictor of microalbuminuria independently of mean HbA_1C_ [11]. Moreover, Lin et al. reported a strong association between HbA_1C_ variability and diabetic nephropathy [12], and Luk [13] and Yang [14] also reported that HbA_1C_ variability could predict the development of CKD and ESRD, respectively, in patients with type 2 DM and preserved renal function.

Most previous studies have excluded patients with DM and the later stages of CKD, and thus little is known about how glycemic control affects the clinical prognosis in these patients. Intensive control of HbA_1C_ to achieve as low a level as possible is controversial in patients with CKD, as this can increase the risk of hypoglycemia due to the prolonged half-life of anti-diabetic drugs, reduced renal insulin clearance, degradation of insulin in peripheral tissues, glycogen storage, and renal gluconeogenesis [15]. Few studies have evaluated the association between HbA_1C_ variability and progression to ESRD in patients with diabetes and moderate to advanced stages of CKD. Accordingly, the aim of this study was to investigate whether HbA_1C_ variability is associated with progression to end-stage renal disease in diabetic patients with stages 3–5 CKD, and whether different stages of CKD affect these associations.

## 2. Results

A total of 388 patients (234 males and 154 females) with type 2 DM and stages 3–5 CKD were enrolled in this study, with a mean age of 65.7 ± 10.9 years. The patients were stratified into three groups according to the tertile of HbA_1C_ SD (<0.43, ≥0.43 to <0.89, and ≥0.89%, respectively). Comparisons of the clinical characteristics of the groups are shown in Table 1. There were 127, 132, and 129 patients in the three groups, respectively. Compared to the patients in tertile 1, those in tertile 3 had higher levels of mean HbA_1C_, serum triglycerides, and total cholesterol.

The median follow-up period was 3.5 (0.5–9.3) years. Of the 388 patients, 108 developed ESRD during the follow-up period, including 105 with hemodialysis and three with peritoneal dialysis. The patients in tertile 3 (vs. tertile 1) were associated with a lower risk of the renal endpoint in the unadjusted model (hazard ratio (HR), 0.493; 95% confidence interval (CI), 0.305–0.796; *p* = 0.004). Figure 1 illustrates the Kaplan–Meier curves for dialysis-free survival (log-rank *p* = 0.012) for all of the patients subdivided according to tertiles of HbA_1C_ SD. The patients in tertile 3 had a better dialysis-free survival than those in tertile 1.

Table 2 shows the HRs of HbA_1C_ SD tertiles for progression to dialysis using the univariate Cox proportional hazards model among different subgroups. The subjects were further divided into two groups based on a mean HbA_1C_ value of 7%. In the patients with an HbA_1C_ level ≥7%, those in tertile 2 (vs. tertile 1; HR, 0.369; 95% CI, 0.167–00.814; *p* = 0.014) and tertile 3 (vs. tertile 1; HR, 0.307; 95% CI, 0.143–0.662; *p* = 0.003) were associated with a lower risk of the renal endpoint in the unadjusted model. However, in the patients with an HbA_1C_ level <7%, HbA_1C_ SD tertiles were not associated with progression to dialysis. Figure 2 illustrates the Kaplan–Meier curves for dialysis-free survival among the subjects with (A) an HbA_1C_ level ≥7% (log-rank *p* = 0.004) and (B) an HbA_1C_ level <7% (log-rank *p* = 0.902).

The patients with an HbA_1C_ level ≥ 7% were further divided into two groups based on the stage of CKD. In the patients with an HbA_1C_ level ≥ 7% and CKD stages 3–4, those in tertile 3 (vs. tertile 1; HR, 0.329; 95% CI, 0.122–0.887; *p* = 0.028) were associated with a lower risk of the renal endpoint in the unadjusted model. However, this relationship was not observed in the patients with an HbA_1C_ level ≥ 7% and CKD stage 5. Figure 3 illustrates the Kaplan–Meier curves for dialysis-free survival among the subjects with (A) an HbA_1C_ level ≥ 7% and CKD stages 3–4 (log-rank *p* = 0.045) and (B) an HbA_1C_ level ≥ 7% and CKD stage 5 (log-rank *p* = 0.808).

Table 3 shows the HR estimates for progression to dialysis with multivariate adjustments in the patients with an HbA_1C_ level ≥ 7% and CKD stage 3–4. The patients in tertile 3 (vs. tertile 1) were associated with a lower risk of the renal endpoint in the adjusted model (HR, 0.243; 95% CI, 0.086–0.688; *p* = 0.008) after adjusting for age, sex, hypertension, coronary artery disease and cerebrovascular disease. This relationship remained significant after further adjustments for mean HbA_1C_, triglycerides, total cholesterol, baseline eGFR, calcium-phosphorous product, uric acid, and the use of angiotensin-converting-enzyme inhibitors (ACEIs) and/or angiotensin II receptor blockers (ARBs) (HR, 0.175; 95% CI, 0.059–0.518; *p* = 0.002).

The patients with an HbA_1C_ level ≥ 7% and CKD stages 3–4 were further divided into two groups based on the trend of HbA_1C_ level (Table 2). In the patients with an HbA_1C_ level ≥ 7%, CKD stages 3–4 and a decreasing HbA_1C_ trend, those in tertile 2 (vs. tertile 1; HR, 0.261; 95% CI, 0.069–0.996; *p* = 0.049) and tertile 3 (vs. tertile 1; HR, 0.245; 95% CI, 0.069–0.869; *p* = 0.029) were associated with a lower risk of the renal endpoint in the unadjusted model. However, in the patients with an HbA_1C_ level ≥ 7%, CKD stages 3–4 and an increasing HbA_1C_ trend, HbA_1C_ SD tertiles were not associated with progression to dialysis. Figure 4 illustrates the Kaplan–Meier curves for dialysis-free survival among the subjects with (A) an HbA_1C_ level ≥ 7%, CKD stages 3–4 and a decreasing HbA_1C_ trend (log-rank *p* = 0.050) and (B) an HbA_1C_ level ≥ 7%, CKD stages 3–4, and an increasing HbA_1C_ trend (log-rank *p* = 0.324).

## 3. Discussion

In the present study, we investigated the association between HbA_1C_ variability and renal outcomes in diabetic patients with stages 3–5 CKD over a follow-up period of 3.5 years. The results showed that the diabetic patients with CKD stages 3–4 and an HbA_1C_ level ≥ 7% in the top HbA_1C_ SD tertile had a decreased risk of progression to dialysis. In contrast, this relationship between HbA_1C_ variability and renal outcome was not significant for those with an HbA_1C_ level < 7% or CKD stage 5.

The most important finding of this study is that greater HbA_1C_ variability was associated with a decreased risk of progression to dialysis in diabetic patients with CKD stages 3–4 and an HbA_1C_ level ≥ 7%. Luk et al. [13] investigated the associations between HbA_1C_ variability and incident CKD and cardiovascular disease in 8439 patients with type 2 diabetic with preserved renal function, and over a follow-up period of 7.2 years found that a high SD of HbA_1C_ was associated with incident CKD and cardiovascular disease, independent of mean HbA_1C_ [13]. Wang et al. [16] also investigated the role of glucose variability, expressed as fluctuations between fasting and 2-h postload glucose in patients with type 2 DM and an HbA_1C_ level ≥ 7%, and found that high short-term glucose variability was associated with decreased eGFR and increased risk of CKD in the patients with poor glycemic control [16]. Furthermore, Yang et al. [14] investigated the relationship between coefficient of variation (CV) of HbA_1C_ and progression to ESRD in patients with type 2 DM during a follow-up period of 8.2 years, and found that high HbA_1C_-CV predicted the development of ESRD. In contrast to these studies [13,14,16], we found that greater HbA_1C_ variability was associated with better, not worse, real outcomes in the diabetic patients with CKD stages 3–4 and an HbA_1C_ level ≥ 7%. To further investigate this inconsistency, we performed subgroup analysis according to HbA_1C_ trend, and found that in the patients with a decreasing HbA_1C_ trend, those with greater HbA_1C_ variability were associated with better real outcomes, but not in those with an increasing HbA_1C_ trend. We hypothesize that patients with a high glucose level may change their hypoglycemic medications or receive more intensive diabetes control, and that this could lead to a decrease in glucose level and higher variation in fasting plasma glucose level. The higher rate of insulin use in patients with tertile 3 may partially support our hypothesis. The United Kingdom Prospective Diabetes Study (UKPDS) trial of patients with type 2 DM and preserved kidney function demonstrated that intensive glycemic control targeting an HbA_1C_ level of < 6–6.5% reduced the development and progression of diabetic nephropathy [17]. Possible mechanisms which may explain the impact of high glucose and renal toxicity include increases in glomerular permeability, circulating levels of inflammatory cytokines, mesangial lipid accumulation, mesangial and tubulointerstitial cell matrix production, expression of fibrinogenesis markers, endothelial dysfunction, and the generation of free radicals that induce diabetic complications [18,19,20,21,22]. Our findings suggest that greater HbA_1C_ variability may be associated with aggressive glucose control, and that this had a positive impact on the renal outcomes of the diabetic patients with stages 3–4 CKD and poor glycemic control (HbA_1C_ ≥ 7%). Our findings would remind physicians the importance of intensive glucose control on renal function progression in CKD patients.

Another important finding of our study is that, in contrast to the patients with CKD stages 3–4, the significance of HbA_1C_ variability and renal outcomes was not observed in those with CKD stage 5. The prognostic role of HbA_1C_ in patients with CKD stage 5 is unclear because of impaired glucose metabolism in patients with advanced CKD, and because HbA_1C_ level can be altered by anemia or the use of erythropoiesis-stimulating agents. A marked reduction in insulin clearance is known to occur until the eGFR falls to < 15–20 mL/min [23]. In addition, the formation of HbA_1C_ is known to be lower in patients with CKD due to a decrease of 30–70% in the lifespan of red blood cells (RBCs), and resistance of carbamylated hemoglobin molecules to glycosylation in a uremic environment [24]. In addition, administering erythropoiesis-stimulating agents to patients with anemia has been shown to augment, the proportion of young RBCs in peripheral blood, and these young RBCs have been shown to have a lower rate of glycosylation than old RBCs, thereby altering the formation of HbA_1C_ [25]. Several studies have reported that in diabetic patients with CKD stages 3–4, a higher HbA_1C_ level (>9%) appeared to be associated with poorer clinical outcomes, but that this was not observed in patients with CKD stage 5 [26,27]. Aggressive glycemic control appears to be beneficial for early diabetic nephropathy, however data supporting intensive glycemic control in patients with advanced CKD (including ESRD) are lacking [28]. Challenges in the management of such patients include therapeutic criteria, monitoring difficulties, and the complexity of management. In summary, HbA_1C_ variability seems to be more useful in predicting clinical outcomes in patients with CKD stages 3–4 than in those with CKD stage 5, possibly because of multiple factors influencing the production of HbA_1C_ or other factors influencing the progression to dialysis in patients with CKD stage 5.

There are several limitations to this study. First, because this was an observational study, the number and frequency of HbA_1C_ measurements varied between individual patients. To minimize the influence of the number and frequency of HbA_1C_ measurements on the results, we excluded patients with fewer than three HbA_1C_ measurements during the follow-up period, and those who were followed for less than six months. However, the lack of uniformity in the number and frequency of HbA_1C_ measurements remains an important limitation of this analysis. In addition, the use of immunoassays in the hospital to test for HbA_1C_ was a limitation, since immunological methods for the detection of HbA_1C_ are more reliable in a uremic environment. Finally, the limited number of study patients severely reduced the power of the study.

## 4. Subjects and Methods

### 4.1. Study Patients and Design

The study was conducted at a regional hospital in southern Taiwan. We consecutively recruited patients with type 2 DM and evidence of kidney damage lasting for more than three months with stages 3–5 CKD according to the National Kidney Foundation-Kidney Disease Outcomes Quality Initiative (K/DOQI) guidelines [29] from January 2007 to September 2015. The patients were classified as having CKD stages 3, 4, and 5, based on an estimated glomerular filtration rate (eGFR) (mL/min/1.73 m^2^) of 30–59, 15–29, and <15, respectively. All of the patients were regularly followed up at our Outpatient Department of Internal Medicine. The exclusion criteria were patients with fewer than three HbA_1C_ measurements during the follow-up period, patients who died, and those who commenced dialysis within six months after enrollment. Finally, 388 patients (mean age 65.7 ± 10.9 years, 234 males) were included in this study. The study protocol was approved by our Institutional Review Board of Kaohsiung Medical University Hospital (KMUHIRB-E (I)-20160032), and all enrolled patients provided written informed consent.

### 4.2. Collection of Demographic, Medical, and Laboratory Data

Demographic data including age and sex and medical data on comorbidities were obtained from medical records or patient interviews. DM was defined as a fasting blood glucose level of >126 mg/dL or the use of hypoglycemic agents to control the level of blood glucose. Hypertension was defined as systolic blood pressure ≥ 140 mmHg or diastolic blood pressure ≥ 90 mmHg or the use of anti-hypertensive medications. Coronary artery disease was defined as evidence of old myocardial infarction, coronary artery disease on angiography, and a history of typical angina with a positive stress test, coronary artery bypass surgery, or angioplasty. Fasting blood samples were obtained from each patient within 1 month of enrollment, and laboratory analyses were performed using an autoanalyzer (Roche Diagnostics GmbH, D-68298 Mannheim COBAS Integra 400, Mannheim, Germany). The compensated Jaffé method was used to measure levels of serum creatinine using a Roche/Integra 400 Analyzer (Roche Diagnostics, Mannheim, Germany) and isotope-dilution mass spectrometry [30]. The four-variable Modification of Diet in Renal Disease equation was used to calculate eGFR [31]. HbA_1C_ was measured using automated cation-exchange high-performance liquid chromatography. Data on the prescriptions of angiotensin converting enzyme inhibitors (ACEIs) and angiotensin II receptor blockers (ARBs) during the study period were obtained from the patients’ medical records.

### 4.3. Serial HbA_1C_ Measurements

HbA_1C_ measurements were recorded for each patient from the date of enrollment until the development of the renal endpoint (see the *Definition of Renal Endpoint* section below) or the censor date (April 2016), whichever occurred first. The mean and standard deviation (SD) of HbA_1C_ were calculated for each patient, with the SD being considered an index of HbA_1C_ variability. The rate of change of HbA_1C_ was assessed using the HbA_1C_ slope, defined as the regression coefficient between HbA_1C_ and time. At least three HbA_1C_ measurements after enrollment were required to estimate the HbA_1C_ slope. An HbA_1C_ slope ≥ 0 was considered to be an increasing trend, whereas an HbA_1C_ slope < 0 was considered to be a decreasing trend.

### 4.4. Definition of Renal Endpoint

The renal endpoint was defined as commencing dialysis. In the patients who reached the endpoint, data on renal function were censored when they commenced renal replacement therapy. All other patients were followed until April 2016. The regulations of the National Health Insurance Bureau of Taiwan were used to confirm the patients who started dialysis therapy, according to laboratory data, nutritional status and symptoms and signs of uremia.

### 4.5. Statistical Analysis

SPSS version 15.0 for Windows (SPSS Inc., Chicago, IL, USA) was used for all statistical analyses. Data are expressed as percentages or mean ± SD. Multiple comparisons among groups were performed using one-way analysis of variance (ANOVA) followed by post hoc tests with Bonferroni correction. The Kaplan–Meier method was used to plot survival curves for the renal endpoint. A Cox proportional hazards model was used to analyze the time to renal endpoint and the risk factors. A *p*-value of <0.05 was considered to indicate a significant difference.

## 5. Conclusions

In conclusion, greater HbA_1C_ variability with a decreasing HbA_1C_ trend, which may be related to intensive diabetes control, was associated with a decreased risk of progression to dialysis in patients with stages 3–4 CKD and poor glycemic control (HbA_1C_ ≥ 7%), but this association was not found in the patients with CKD stage 5. Our results support the potential role of aggressive glycemic control on clinical outcomes and highlight its importance in diabetic patient with stages 3–4 CKD.

## Figures and Tables

**Figure 1 ijms-19-04116-f001:**
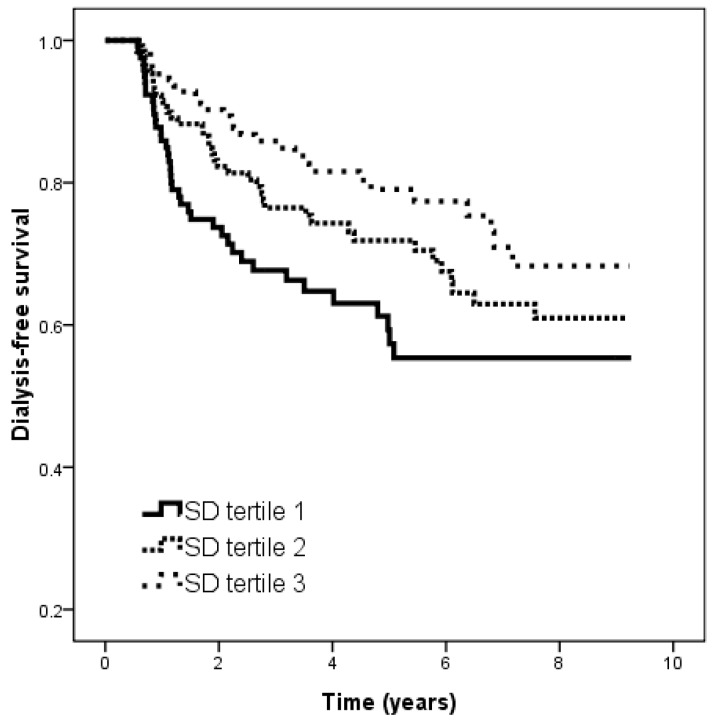
Kaplan–Meier analysis of dialysis-free survival according to tertiles of HbA_1C_ standard deviation (SD) (log-rank *p* = 0.012). Patients with tertile 3 of HbA_1C_ SD had a better renal-free survival than those with tertile 1 of HbA_1C_ SD.

**Figure 2 ijms-19-04116-f002:**
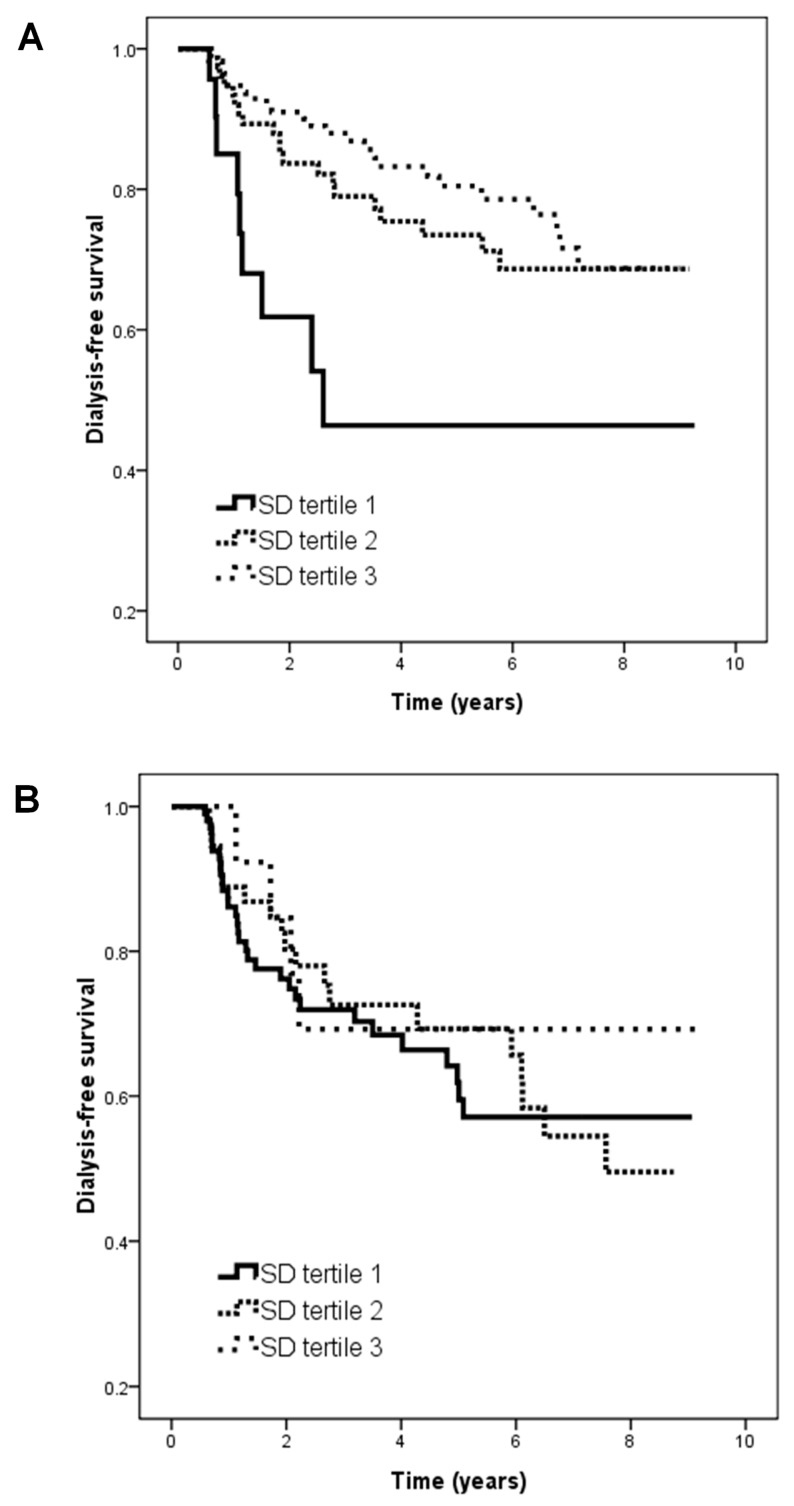
Kaplan–Meier analysis of dialysis-free survival according to tertiles of HbA_1C_ SD among subjects with (**A**) HbA_1C_ ≧ 7% (log-rank *p* = 0.004) and (**B**) HbA_1C_ < 7% (log-rank *p* = 0.902).

**Figure 3 ijms-19-04116-f003:**
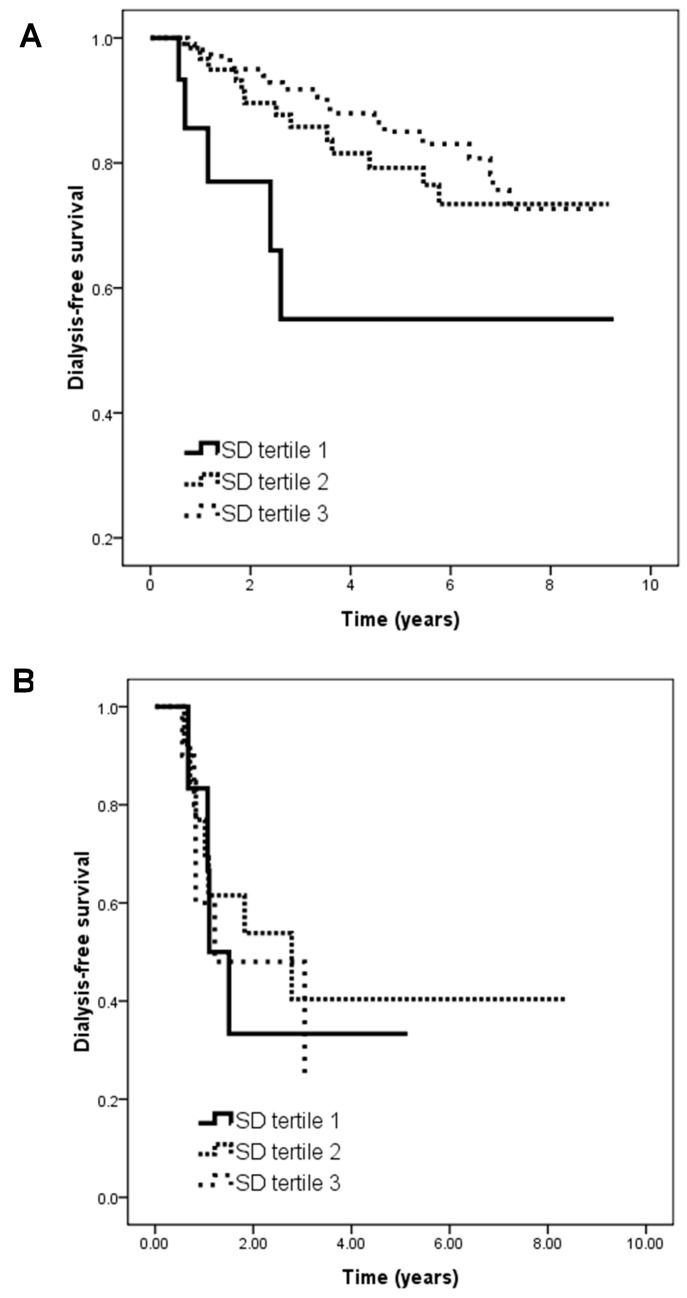
Kaplan–Meier analysis of dialysis-free survival according to tertiles of HbA_1C_ SD among subjects with (**A**) HbA_1C_ ≧ 7% and CKD stages 3–4 (log-rank *p* = 0.045) and (**B**) HbA_1C_ ≧ 7% and CKD stage 5 (log-rank *p* = 0.808).

**Figure 4 ijms-19-04116-f004:**
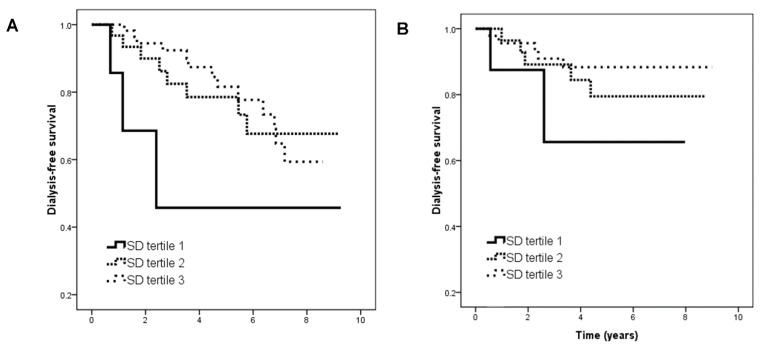
Kaplan–Meier analysis of dialysis-free survival according to tertiles of HbA_1C_ SD among subjects with (**A**) HbA_1C_ ≧ 7%, CKD stages 3–4 and downward HbA_1C_ trend (log-rank *p* = 0.050) and (**B**) HbA_1C_ ≧ 7%, CKD stages 3–4 and upward HbA_1C_ trend (log-rank *p* = 0.324).

**Table 1 ijms-19-04116-t001:** Comparison of clinical characteristics according to tertiles of standard deviation (SD) of HbA_1C_.

Characteristics	SD Tertile 1 (*n* = 127)	SD Tertile 2 (*n* = 132)	SD Tertile 3 (*n* = 129)	*p*
SD (%)	0.2 ± 0.1	0.7 ± 0.1 *	1.5 ± 0.6 *^,†^	<0.001
Age (year)	66.1 ± 106	66.3 ± 10.8	64.7 ± 11.4	0.433
Male gender (%)	65.4	59.8	55.8	0.295
Hypertension (%)	93.7	90.2	93.8	0.444
Coronary artery disease (%)	12.6	12.2	13.3	0.967
Cerebrovascular disease (%)	15.0	12.1	13.2	0.797
CKD Stage				0.096
Stage 3 (%)	33.9	40.2	35.7	
Stage 4 (%)	35.4	36.4	52.7	
Stage 5 (%)	30.7	22.7	11.6	
Smocking (%)	36.2	24.2	35.5	0.091
Systolic blood pressure (mmHg)	146.0 ± 19.0	148.6 ± 21.0	144.1 ± 20.1	0.194
Diastolic blood pressure (mmHg)	74.8 ± 12.7	76.1 ± 11.7	75.3 ± 12.0	0.656
Body mass index (kg/m^2^)	25.3 ± 3.6	26.2 ± 4.0	26.7 ± 4.2 *	0.021
Laboratory Parameters				
Hemoglobin (g/dL)	11.1 ± 2.1	11.3 ± 2.0	11.5 ± 2.0	0.243
Mean HbA_1C_ (%)	6.6 ± 0.9	7.4 ± 1.0^*^	8.6 ± 1.4 *^,†^	<0.001
HbA_1C_ measurement frequency (times)	5.5 ± 4.5	9.6 ± 6.4*	9.7 ± 5.9 *	<0.001
Triglyceride (mg/dL)	156.9 ± 85.2	181.8 ± 109.7	216.9 ± 109.7 *^,†^	<0.001
Total cholesterol (mg/dL)	189.1 ± 46.2	194.4 ± 57.8	208.9 ± 60.6 *	0.013
Baseline eGFR (mL/min/1.73 m^2^)	25.4 ± 13.5	27.7 ± 13.9	27.7 ± 11.2	0.263
Calcium-phosphorous product (mg^2^/dL^2^)	38.0 ± 8.1	37.8 ± 7.3	37.7 ± 5.6	0.923
Uric acid (mg/dL)	8.3 ± 2.2	8.1 ± 1.9	8.1 ± 2.0	0.800
Medications				
ACEI and/or ARB use (%)	66.9	75.8	73.6	0.258
Statin	32.3	34.8	48.8 *	0.013
Oral antidiabetic agent	83.3	86.6	89.1	0.413
Insulin	17.3	26.8	37.5 *	0.001
Erythropoetin	23.6	25.0	17.8	0.339

Abbreviations. SD, standard deviation; CKD, chronic kidney disease; eGFR, estimated glomerular filtration rate; ACEI, angiotensin converting enzyme inhibitor; ARB, angiotensin II receptor blocker. The study patients were stratified into 3 groups according to tertiles of SD of HbA_1C_ (<0.43, ≧0.43–<0.89, and ≧0.89%). * *p* < 0.05 compared to patients with tertile 1 of SD of HbA_1C_, ^†^
*p* < 0.05 compared to patients with tertile 2 of SD of HbA_1C_.

**Table 2 ijms-19-04116-t002:** Relation of SD tertile of HbA_1C_ for progression to dialysis using univariate Cox proportional hazards model among different subgroups.

SD of HbA_1C_	Unadjusted	
	Hazard Ratios (95% CI)	*p*
All patients (*n* = 388)		
Tertile 1 (*n* = 127)	1	
Tertile 2 (*n* = 132)	0.680 (0.437–1.058)	0.088
Tertile 3 (*n* = 129)	0.493 (0.305–0.796)	0.004
HbA_1C_ ≧ 7% (*n* = 218)		
Tertile 1 (*n* = 25)	1	
Tertile 2 (*n* = 77)	0.369 (0.167–0.814)	0.014
Tertile 3 (*n* = 116)	0.307 (0.143–0.662)	0.003
HbA_1C_ < 7% (*n* = 170)		
Tertile 1 (*n* = 102)	1	
Tertile 2 (*n* = 55)	0.940 (0.530–1.666)	0.832
Tertile 3 (*n* = 13)	0.793 (0.280-2.249)	0.663
HbA_1C_ ≧ 7% and CKD stages 3–4 (*n* = 185)		
Tertile 1 (*n* = 17)	1	
Tertile 2 (*n* = 62)	0.378 (0.134–1.064)	0.065
Tertile 3 (*n* = 106)	0.329 (0.122–0.887)	0.028
HbA_1C_ ≧ 7% and CKD stage 5 (*n* = 33)		
Tertile 1 (*n* = 8)	1	
Tertile 2 (*n* = 15)	0.746 (0.217–2.562)	0.642
Tertile 3 (*n* = 10)	1.046 (0.294–3.720)	0.945
HbA_1C_ ≧ 7%, CKD stages 3–4 and HbA_1C_ downward trend (*n* = 98)		
Tertile 1 (*n* = 7)	1	
Tertile 2 (*n* = 32)	0.261 (0.069–0.996)	0.049
Tertile 3 (*n* = 59)	0.245 (0.069–0.869)	0.029
HbA_1C_ ≧ 7%, CKD stages 3–4 and HbA_1C_ upward trend (*n* = 87)		
Tertile 1 (*n* = 10)	1	
Tertile 2 (*n* = 30)	0.459 (0.088–2.295)	0.356
Tertile 3 (*n* = 47)	0.315 (0.062–1.596)	0.163

Values expressed as Hazard Ratios and 95% confidence interval (CI).

**Table 3 ijms-19-04116-t003:** Relation of SD tertile of HbA_1C_ for progression to dialysis using multivariate stepwise Cox proportional hazards model in patients with HbA_1C_ ≧ 7% and CKD stages 3–4.

Parameters	Multivariate Adjusted (1)		Multivariate Adjusted (2)	
	Hazard Ratios (95% CI)	*p*	Hazard Ratios (95% CI)	*p*
SD of HbA_1C_				
Tertile 1	1		1	
Tertile 2	0.367 (0.130–1.038)	0.059	0.398 (0.134–1.179)	0.096
Tertile 3	0.243 (0.086–0.688)	0.008	0.175 (0.059–0.518)	0.002

Values expressed as Hazard Ratios and 95% confidence interval (CI). Multivariate model (1): adjusted for age, sex, hypertension, coronary artery disease and cerebrovascular disease. Multivariate model (2): model (1) + mean HbA_1C_, triglyceride, total cholesterol, baseline eGFR, calcium-phosphorous product, uric acid and ACEI and/or ARB use.

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
