# Peer review of "Association of HbA1C Variability and Renal Progression in Patients with Type 2 Diabetes with Chronic Kidney Disease Stages 3–4"

_ijms, 2018, doi:10.3390/ijms19124116_

Reviewer 1 Report

Lee et al., performed a longitudinal study in 388 diabetic patients with chronic kidney disease stage 3-5 with an aim to investigate whether HbA1c variability is associated with progression to end-stage renal disease. The authors found that greater HbA1c variability was associated with a lower risk of progression to dialysis in the patients with stage 3-4 CKD and poor glycemic control (HbA1c ≥ 7%)

The authors have done an excellent job in planning the study. The methods are robust, results are well described and provided a great discussion.

I have the following comments to the authors,

Introduction:

-Though mentioned in abstract, please include a brief paragraph describing what is ‘Glucose or Glycemic variability’ with appropriate references, to help the naïve readers.

Glucose variability could be defined in several ways, within-day variability, between-day variability, and long-term variability expressed using changes in HgbA1C, (1), which the authors used.

Subjects and methods:

-As the Hgb A1c is affected by the hemolysis or anemia, those on iron, vitamin B12, or folate or deficiency, and patients on erythropoietin did the authors exclude such patients in the study?

-The iron deficiency and being on Erythropoietin becomes increasingly common with advanced CKD especially with eGFR less than 25. Does this influence any of the findings in CKD 5 patients in the study?

Also, were any of the patients with abnormal hemoglobinopathies excluded?

Discussion:

- Highly variable A1C was found be associated with some factors in patients such as increased smoking, higher blood pressure, and an increased prevalence of peripheral neuropathy, peripheral vascular disease and might be indicative of unhealthy lifestyle behavior, non-compliance, comorbidities, or complications, (2,3,4)

Have the authors noticed any such practices or findings in their study participants?

 -Can authors comment on the use of fructosamine or Glycated albumin (GA) which are not affected by hemoglobin levels and potentially could be superior to HbA1C, in patients with advanced CKD who suffer from iron deficiency, anemia and need of the erythropoietin? Also, GA is not influenced by the serum albumin levels which could be low in diabetic nephropathy patients.

Thank you for submitting this exciting study to IJMS.

References:

1. Cavalot F (2013) Do data in the literature indicate that glycaemic variability is a clinical problem? Glycaemic variability and vascular complications of diabetes. Diabetes ObesMetab15: S3–S8

2.Wadén J, Forsblom C, Thorn LM, Gordin D, Saraheimo M, Groop PH, Finnish Diabetic Nephropathy Study Group.Diabetes. 2009 Nov; 58(11):2649-55.

3. Sugawara A, Kawai K, Motohashi S, Saito K, Kodama S, Yachi Y, Hirasawa R, Shimano H, Yamazaki K, Sone H, Diabetologia. 2012 Aug; 55(8):2128-31.

4. Lin CC, Chen CC, Chen FN, Li CI, Liu CS, Lin WY, Yang SY, Lee CC, Li TC. Am J Med. 2013 Nov; 126(11): 1017.e1-10.

Author Response

Response to Reviewer 1 Comments

Comments and Suggestions for Authors

Lee et al., performed a longitudinal study in 388 diabetic patients with chronic kidney disease stage 3-5 with an aim to investigate whether HbA1c variability is associated with progression to end-stage renal disease. The authors found that greater HbA1c variability was associated with a lower risk of progression to dialysis in the patients with stage 3-4 CKD and poor glycemic control (HbA1c ≥ 7%)

The authors have done an excellent job in planning the study. The methods are robust, results are well described and provided a great discussion.

I have the following comments to the authors,

Point 1: Introduction:

-Though mentioned in abstract, please include a brief paragraph describing what is ‘Glucose or Glycemic variability’ with appropriate references, to help the naïve readers.

Glucose variability could be defined in several ways, within-day variability, between-day variability, and long-term variability expressed using changes in HgbA1C, (1), which the authors used.

Response 1: Thank you for your prestigious comments. We added a paragraph to describe glycemic variability in the part of introduction from your suggested reference.

Point 2: Subjects and methods:

-As the Hgb A1c is affected by the hemolysis or anemia, those on iron, vitamin B12, or folate or deficiency, and patients on erythropoietin did the authors exclude such patients in the study?

-The iron deficiency and being on Erythropoietin becomes increasingly common with advanced CKD especially with eGFR less than 25. Does this influence any of the findings in CKD 5 patients in the study?

Also, were any of the patients with abnormal hemoglobinopathies excluded?

Response 2: As your comments, we included the data of EPO users in Table 1. We also stated the Hgb level of each tertile to describe the status of hemoglobinopathies.

Point 3: Discussion

- Highly variable A1C was found be associated with some factors in patients such as increased smoking, higher blood pressure, and an increased prevalence of peripheral neuropathy, peripheral vascular disease and might be indicative of unhealthy lifestyle behavior, non-compliance, comorbidities, or complications, (2,3,4)

Have the authors noticed any such practices or findings in their study participants?

Response 3: As your comments, we included the history of smoking, the systolic and diastolic blood pressure in Table 1, which are factors that might influence the high Hba1C variability.

Point 4: Discussion: 

-Can authors comment on the use of fructosamine or Glycated albumin (GA) which are not affected by hemoglobin levels and potentially could be superior to HbA1C, in patients with advanced CKD who suffer from iron deficiency, anemia and need of the erythropoietin? Also, GA is not influenced by the serum albumin levels which could be low in diabetic nephropathy patients.

Response 4: Thank you for your reminder. Although Hb A1C is not recommended in clinical situations which may interfere with the metabolism of hemoglobin, such as in hemolytic, secondary or iron deficiency anemia, hemoglobinopathies, pregnancy, and uremia. And glycated albumin (GA) is higher glycated portion of fructosamine, which is a test that reflects short-term glycemia and is not influenced by situations that falsely alter A1C levels. However, our study participants’ Hgb levels were still above 11gm/dl, which Hba1c is still an appropriate test for evaluating their glucose control, so we chose HbA1C over GA, but we consider GA as test in our future study.

Reviewer 2 Report

The authors describe the influence of individual HbA1C variability on the progression CKD to renal failure in 388 patients (stage 3-5 CKD) with type 2 diabetes over a follow-up period of 9 years (median 3.5)

Their main finding is that a high variability in HbA 1c level was associated with a decreased risk of progression to dialysis in diabetic patients with CKD stage 3-4 and HbA 1c levels smaller than7%.

They suggest that an aggressive glucose control in these patients is reposible for a greater variability in HbA 1c and that this may have had a positive impact on the renal outcome.

This is an interesting new hypothesis. Nevertheless, the presented data of this retrospective study (with all immanent limitations) is in contradiction to clinical expectations and most of the published work. Therefore, it is of particular importance to exclude all sources of errors and all relevant data must be taken into account.

In my opinion the study has some serious flaws:

No Data were presented about the frequency of the HbA 1c measurements and if they are performed on a regular or arbitrary scheme. According to the manuscript the minimal number of HbA 1c determinations was 3. This is not sufficient for a maximal observation period of nine years. Detailed data about frequency and the intervals between HbA 1c measurements should be provided and analysed.

Since inter-individual differences in anti-hyperglycemic therapy is the key approach to the interpretation of the results, data about the therapy must be included, especially since a high variation of HbA1c is commonly associated with a non-optimal therapy.

Minor:

Table 2 the number of cases in the various sub-groups should be provided.

Table 1: Since ARB are also used in the therapy of cardiac insufficiency and insufficient adjustment of blood pressure contributes to the progression of CKD, more detailed data about blood pressure  would be desirable.

Author Response

Response to Reviewer 2 Comments

Comments and Suggestions for Authors

The authors describe the influence of individual HbA1C variability on the progression CKD to renal failure in 388 patients (stage 3-5 CKD) with type 2 diabetes over a follow-up period of 9 years (median 3.5)

Their main finding is that a high variability in HbA 1c level was associated with a decreased risk of progression to dialysis in diabetic patients with CKD stage 3-4 and HbA 1c levels smaller than7%.

They suggest that an aggressive glucose control in these patients is reposible for a greater variability in HbA 1c and that this may have had a positive impact on the renal outcome.

This is an interesting new hypothesis. Nevertheless, the presented data of this retrospective study (with all immanent limitations) is in contradiction to clinical expectations and most of the published work. Therefore, it is of particular importance to exclude all sources of errors and all relevant data must be taken into account.

In my opinion the study has some serious flaws:

Point 1: No Data were presented about the frequency of the HbA 1c measurements and if they are performed on a regular or arbitrary scheme. According to the manuscript the minimal number of HbA 1c determinations was 3. This is not sufficient for a maximal observation period of nine years. Detailed data about frequency and the intervals between HbA 1c measurements should be provided and analysed.

Thank you for your important reminder. We now state the frequency and intervals of Hba1c measurement in Table 1.

Point 2: Since inter-individual differences in anti-hyperglycemic therapy is the key approach to the interpretation of the results, data about the therapy must be included, especially since a high variation of HbA1c is commonly associated with a non-optimal therapy.

As your comments, we now state the user of oral antidiabetic agent and insulin in each tertile at Table 1.

Point 3: Table 2 the number of cases in the various sub-groups should be provided.

As your comments, we now state the number of cases in the various sub-groups at Table 2.

Point 4: Table 1: Since ARB are also used in the therapy of cardiac insufficiency and insufficient adjustment of blood pressure contributes to the progression of CKD, more detailed data about blood pressure  would be desirable.

: As your comments, we included the data of systolic and diastolic blood pressure of each tertile in Table 1.

Reviewer 3 Report

The study describes nicely the association between HbA1C variability and CKD ( and progression to the ESRD). I would include additional information :

The title can be mis-interpreted; the found association between higher HbA1C variability and decreased risk for CKD progression may be the result of a most intensive plasma glucose control, as suggested by the latest analysis of the study.  Please provide a clearer message in the title;

I would include details on diabetes medications (number of drugs, insulin vs oral diabetic drugs etc) and BMI of patients.

Since often diabetes patients are obese, how did you estimate the GFR?

Author Response

Response to Reviewer 3 Comments

Comments and Suggestions for Authors

The study describes nicely the association between HbA1C variability and CKD ( and progression to the ESRD). I would include additional information:

Point 1: The title can be mis-interpreted; the found association between higher HbA1C variability and decreased risk for CKD progression may be the result of a most intensive plasma glucose control, as suggested by the latest analysis of the study.  Please provide a clearer message in the title;

If we are not misinterpreting your comment, do you want us to revise our title to “Intensive Glucose Control Leading to Greater HbA1C Variability is Associated with a Decreased Risk of Progression to Dialysis in Patients with Type 2 Diabetes with Chronic Kidney Disease Stage 3-4”.

Point 2: I would include details on diabetes medications (number of drugs, insulin vs oral diabetic drugs etc) and BMI of patients.

Thank you for comments. We included the details of diabetes medication including oral antidiabetic agents and insulin in Table 1. BMI was also stated.

Point 3: Since often diabetes patients are obese, how did you estimate the GFR?

Response to Reviewer: We calculated our eGFR by using MDRD formula.

Round  2

Reviewer 2 Report

The authors have added the appropriate data. Since the lowest number of HbA1c measurements (associated with an expected higher SD), the data does not contradict their conclusions.

It appears that patients in tertile 3 were more frequently treated with insulin and statins. The authors should comment on that in the discussion. 

Author Response

Response to Reviewer 2 Comments

Comments and Suggestions for Authors

The authors have added the appropriate data. Since the lowest number of HbA1c measurements (associated with an expected higher SD), the data does not contradict their conclusions.

Ans: Thank you for your comments.

It appears that patients in tertile 3 were more frequently treated with insulin and statins. The authors should comment on that in the discussion. 

Ans: Thank you for your comments. We add some sentences in the first paragraph of the Discussion (labelled with underline, line 182-193 of the manuscript).

* The higher rate of insulin use in patients with tertile 3 may partially support our hypothesis. The UKPDS trial of patients with type 2 DM and preserved kidney function demonstrated that intensive glycemic control targeting an HbA1c level of < 6%-6.5% reduced the development and progression of diabetic nephropathy [17]. Possible mechanisms which may explain the impact of high glucose and renal toxicity include increases in glomerular permeability, circulating levels of inflammatory cytokines, mesangial lipid accumulation, mesangial and tubulointerstitial cell matrix production, expression of fibrinogenesis markers, endothelial dysfunction, and the generation of free radicals that induce diabetic complications [18-22]. Our findings suggest that greater HbA1c variability may be associated with aggressive glucose control, and that this had a positive impact on the renal outcomes of the diabetic patients with stage 3-4 CKD and poor glycemic control (HbA1c ³ 7%). Our findings would remind physicians the importance of intensive glucose control on renal function progression in CKD patients.